# Adoption of evidence-based medicine: A comparative study of hospital and community pharmacists in Saudi Arabia

**Fahad Alzahrani**[ID][1*], **Nawaf Almutairi**[ID][2], **Abdullah Aloufi**[ID][1], **Abdulmalik Kattan**[3], **Abdulaziz Hakeem**[3], **Mohammed Alharbi**[3], **Naif Alarawi**[3], **Haifa A. Fadil**[ID][1], **Ehsan Habeeb**[1]

**1** Department of Pharmacy Practice, Faculty of Pharmacy, Taibah University, Madinah, Saudi Arabia, **2** King Faisal Specialist Hospital and Research Centre, Jeddah, Saudi Arabia, **3** PharmD Graduate, College of Pharmacy Taibah University, Madinah, Saudi Arabia

\* fzahrani@taibahu.edu.sa

## Abstract

### Objectives

Evidence-based medicine (EBM) combines clinical expertise, patient values, and the best available evidence to guide healthcare decision-making. Despite its importance in pharmacy practice, EBM adoption in Saudi Arabian pharmacies remains under-researched. This study aimed to assess the knowledge, attitudes, and practices regarding EBM among hospital and community pharmacists in the Madinah Region, Saudi Arabia.

### Methods

A cross-sectional study was conducted with 206 pharmacists from September to November 2023. Data were collected through a validated online self-administered questionnaire to evaluate pharmacists' knowledge, attitude, and practice (KAP), as well as their understanding of EBM technical terms.

### Ethical approval

The study was approved by the Scientific Ethics Committee of the College of Pharmacy at Taibah University, Madinah region, Saudi Arabia (reference number COPTU-REC-77–20230827). All participants received a consent form before participating.

### Results

Pharmacists demonstrated moderate knowledge (76.5%), neutral attitudes (76%), and fair practices (68%) toward EBM, with hospital pharmacists scoring higher than community pharmacists. Moreover, 83.3% believed that EBM could enhance patient health outcomes, 80.0% were willing to learn, and 35.9% believed that EBM focuses solely on research without considering clinical experience. Time constraints (34.0%) were a major barrier, and 46.1% of the participants lacked appropriate training. EBM education

**Data availability statement:** All relevant data are within the article and its Supporting Information files. The datasets necessary to replicate the study findings and the supporting files have also been deposited in the Zenodo public data repository. The data can be accessed at the following DOI: 10.5281/zenodo.15332654.

**Funding:** The author(s) received no specific funding for this work.

**Competing interests:** The authors have declared that no competing interests exist.

was correlated with higher knowledge and attitude scores; however, it had no significant impact on practice scores. Significant barriers identified were the difficulty in conveying technical terms (16%) and limited access to adequate training opportunities.

## Conclusion

Despite positive attitudes toward EBM, many pharmacists perceive it as a potential threat to good clinical practice. This perception underscores the need for targeted educational initiatives that promote EBM benefits, address misconceptions, and provide practical support for its integration in both hospital and community pharmacy settings.

## Introduction

Over the past decade, evidence-based medicine (EBM) has gained significant attention from healthcare professionals [1]. EBM integrates scientific evidence, clinical expertise, and patient values to improve medical decision-making [2,3].

Traditionally, pharmacy practice has focused on the dispensing of medications, both prescription and over the counter, with pharmacists providing drug-focused services. However, pharmaceutical care shifts this focus toward a more collaborative pharmacist-patient relationship, actively involving the pharmacist in the treatment process [4,5].

EBM in pharmacy practice is essential to deliver pharmaceutical care, as demonstrated by multiple studies showing its significant impact on clinical decision-making accuracy and improved patient outcomes [6,7]. Research has shown that implementing EBM can lead to more rational drug use, enhance patient safety, and reduce medication errors [8,9].

However, adopting an EBM is not without challenges. Pharmacists and other healthcare professionals often encounter barriers such as time constraints, limited access to resources, difficulties in understanding statistical terminology, and gaps in the knowledge and skills needed to search for and appraise evidence [10,11]. Moreover, the application of EBM must be balanced with individual patient circumstances and preferences.[12].

Saudi Arabia has examined the perceptions, attitudes, and use of EBM among healthcare professionals, including pharmacists. These studies consistently revealed a positive attitude toward EBM, with professionals recognizing its potential to enhance patient care. Nonetheless, significant organizational, professional, and interprofessional barriers continue to hinder the widespread implementation [11,13,14].

Hospital pharmacists typically work in structured environments that foster professional development and facilitate the integration of EBM into their practice. They often collaborate within multidisciplinary teams, which enhances opportunities for knowledge sharing and the practical application of EBM principles [15]. In contrast, community pharmacists face distinct challenges, such as high workloads and limited access to continuing education programs, which can hinder their ability to incorporate EBM into daily practice [16,17]. Chun and Anwer added that, unlike their hospital counterparts, community pharmacists often work independently, necessitating tailored strategies and targeted support to adopt EBM in their professional routines effectively [18].

While previous research has examined EBM adoption among hospital pharmacists in Saudi Arabia [13,19], there is a significant lack of studies investigating EBM implementation in community pharmacy settings. This significant research gap is particularly concerning given that community pharmacists serve as primary healthcare providers for many patients [20]. Furthermore, no studies have directly compared EBM knowledge, attitudes, and practices between hospital and community pharmacists, limiting our understanding of how different practice settings influence evidence-based practice implementation. To bridge this gap, this study aimed to address the existing research gap by comprehensively assessing the knowledge, attitudes, and practices regarding EBM among hospital and community pharmacists in the Madinah region of Saudi Arabia.

## Methods

### Study design and settings

A cross-sectional study was conducted among community and hospital pharmacists in the Madinah region of Saudi Arabia from September to November 2023 to collect data on the knowledge, attitudes, and practices of EBM using a self-reported online questionnaire created with Google Forms.

The study included licensed community and hospital pharmacists with a bachelor's degree or higher who were working full-time in the Madinah region of Saudi Arabia. Pharmacists employed in other sectors, such as pharmaceutical companies, manufacturing, or academia, were excluded as they are typically not involved in direct patient care or clinical decision-making where evidence-based medicine is applied. Moreover, those practicing outside the specified region, pharmacy technicians, individuals who declined to participate, or respondents who did not complete the full survey were also excluded from the study.

### Study sampling and response rate

Hospital pharmacists were recruited using convenience sampling from various government and private hospitals across the Madinah region. Eight trained research assistants obtained recruitment and consent in person, having received comprehensive training on standardized data collection protocols and ethical research practices before study commencement.

For community pharmacists, convenience sampling was the primary recruitment strategy, supplemented by snowball sampling. The study was promoted through internal distribution lists at several local independent and chain pharmacies, and participants were encouraged to share the invitation with eligible colleagues. This approach was necessary due to the absence of a comprehensive and accessible sampling frame for pharmacists in the Madinah region.

A total of 540 pharmacists were invited to participate in the study, of whom 276 completed the questionnaire in whole or in part, yielding an initial response rate of 51.11%. After excluding incomplete and ineligible responses, 206 valid questionnaires were retained for analysis, resulting in a final valid response rate of 38.15%.

### Data collection

An online validated questionnaire was used for data collection, which the researchers developed based on the Noor EBM questionnaire [21]. The questions on EBM barriers and practices were adapted with input from experts and relevant literature [22–24]. The questionnaire was reviewed by three academics and two community pharmacists and piloted for clarity. Pharmacy graduates (n = 32) completed the survey twice within 30 minutes to 1 hour, with score stability tested using the test-retest method. Pearson's correlation showed significant score stability (r > 0.94, p < 0.01), exceeding the acceptable threshold of 80% [25]. Internal consistency was confirmed with a Cronbach's alpha of 91.2% and domain-specific alphas of 90.0%, 93.0%, and 91.0% for EBM knowledge, attitude, and practice, respectively.

The survey contained 40 questions across four areas: (1) professional traits, (2) self-assessed EBM knowledge, (3) attitudes and actions toward EBM, and (4) experiences with EBM, mainly statistical terms. Responses were collected using a

5-point Likert scale and grouped into Agree, Neutral, and Disagree categories. Net agreement scores were calculated by subtracting disagreement from agreement percentages, ranging from -100% to +100%, where positive scores indicated agreement. Similarly, a net frequency score was computed by subtracting low-frequency ("Never" and "Rarely") from high-frequency ("Frequently" and "Very Frequently") responses, with positive values reflecting higher reported behavior frequency. Comparable net scoring methods have been employed in survey research to efficiently summarize ordinal responses and enable meaningful comparisons across groups or time periods [26,27].

The total scores for each section—knowledge, attitude, and practice—were converted into percentage scores by dividing the raw scores by the maximum possible score and multiplying by 100. Classification was based on Bloom's cut-off points, a widely used method in KAP studies. Scores of 60–79% were categorized as moderate knowledge, neutral attitudes, and adequate EBM practices, while scores above 79% indicated excellent knowledge, positive attitudes, and proficient practices. Scores below 60% reflected limited knowledge, negative attitudes, and poor practices [28]. This approach provides a standardized and interpretable framework for assessing EBM-related competencies, consistent with previous health research [29–31].

### Data analysis

Data collected via Google Forms was transferred to an Excel spreadsheet and coded for statistical analysis. IBM SPSS Statistics version 27 was used for the analysis. Descriptive statistics, including means, standard deviations, total scores, frequencies, and percentages, were calculated. Due to non-normal data distribution, non-parametric tests were used. The Mann-Whitney U test and the Kruskal-Wallis test were applied, followed by Dunn's test for multiple pairwise comparisons. The significance level ($\alpha$) for all statistical tests was set at 0.05, and two-tailed tests were employed throughout the analysis. Effect size was evaluated using Cohen's d. Based on Cohen's guidelines, an effect size of 0.2 indicates a small effect, 0.5 a medium effect, and 0.8 or greater a large effect. A small effect ($d \approx 0.2$) reflects a modest difference between groups that, although potentially statistically significant, may have limited practical relevance [32].

## Results

### Pharmacists' characteristics

The mean age was comparable between hospital ($31.58 \pm 5.4$ years) and community pharmacists ($31.06 \pm 5.15$ years). Gender distribution showed significant variation, with community pharmacies having predominantly male practitioners (90.0%) compared to hospitals showing more gender diversity (56.6% male, 43.4% female). Educational qualifications differed markedly between settings. Hospital pharmacists demonstrated higher academic achievements, with 44.3% holding PharmD degrees. In contrast, community pharmacists primarily held B-Pharm degrees (81.0%).

The source of education emerged as a distinctive factor; 92.5% of hospital pharmacists graduated from Saudi universities, while 67.0% of community pharmacists received foreign education. Experience distribution revealed that half of hospital pharmacists (50.0%) had 0–5 years of practice, whereas community pharmacists showed a more even distribution across experience levels, with 37.0% having over 10 years of experience. Regarding EBM training, hospital pharmacists reported higher participation rates (59.5%) compared to community pharmacists (47.0%). The comprehensive demographic and professional characteristics of the study pharmacists are detailed in Table 1.

### Knowledge of EBM

Table 2. highlights pharmacists' knowledge and perceptions of EBM. Most (84.4%) agreed that EBM involves critically appraising research for clinical decision-making. However, 41.7% of the pharmacists believed EBM focuses solely on research without considering clinical experience, with a low net agreement of 5.8%. Approximately 47.0% prioritized patient preferences over clinician preferences, with a net agreement of 25.2%. A significant majority of pharmacists (81.1%) felt that EBM improved their understanding of research methodology, with a high net agreement of 79.7%. EBM's

**Table 1. Pharmacists' demographic and professional characteristics.**

| Characteristics | Hospital pharmacists | Community pharmacists |
|---|---|---|
| | n (%) | n (%) |
| **Age, years n (SD)** | 31.58±5.4 | 31.06±5.15 |
| **Gender** | | |
| Male | 60 (56.6) | 90 (90.0) |
| Female | 46 (43.4) | 10 (10.0) |
| **Level of Education** | | |
| Bachelor's Degree in Pharmacy (B-Pharm) | 46 (43.4) | 81 (81.0) |
| Bachelor's Degree in Doctor of Pharmacy (PharmD) | 47 (44.3) | 16 (16.0) |
| Postgraduate (MSc, PhD) in Pharmacy | 13 (12.3) | 3 (3.0) |
| **Source of Education** | | |
| Saudi University | 98 (92.5) | 33 (33.0) |
| Foreign University | 7 (7.5) | 67 (67.0) |
| **Years of Experience** | | |
| 0–5 | 53 (50.0) | 39 (39.0) |
| 6–10 | 29 (28.3) | 24 (24.0) |
| >10 | 23 (21.6) | 37 (37.0) |
| **Attended an Education/Training Program on EBM** | | |
| Yes | 63 (59.5) | 47 (47.0) |
| No | 43 (40.5) | 53 (53.0) |

**Table 2. Pharmacists' knowledge of EBM.**

| Statement | SA/A | N | SD/D | Positive responses | Negative responses | Net agreement |
|---|---|---|---|---|---|---|
| EBM involves the process of critically appraising research findings as the basis for making the best clinical decision | 174 | 32 | 0 | 84.4% | 0.0% | 84.4% |
| EBM focuses on the best currently available research without considering clinical experience. | 86 | 46 | 74 | 41.7% | 35.9% | 5.8% |
| Patients' preferences should be prioritized over clinicians' preferences in making clinical decisions | 97 | 64 | 45 | 47.0% | 21.8% | 25.2% |
| EBM improves clinicians' understanding of research methodology | 168 | 35 (17) | 3 | 81.1% | 1.4% | 79.7% |
| EBM can be practiced in situations where there is doubt about any aspect of clinical management | 151 | 47 | 8 | 73.3 | 3.9% | 69.4% |
| The increasing number of systematic reviews that are applicable to practice can be found in the Cochrane Library | 122 | 79 | 5 | 59.9% | 2.4% | 57.5 |
| Difficulty in understanding statistical terms is the major setback in applying evidence-based medicine | 135 | 59 | 11 | 65.5% | 5.3% | 60.2% |
| Total Score | | Means (%) | | | General Impression | |
| 5444/7210 | | 26.5 (76.5) | | | Moderate Knowledge | |

SD = Strongly Disagree, D = Disagree, N = Neutral, A = Agree, SA = Strongly Agree.

Net Agreement = (% Agree + % Strongly Agree) - (% Strongly Disagree + % Disagree.

applicability in clinical uncertainty was acknowledged by 73.3%, and 59.9% recognized the Cochrane Library as a key resource. Additionally, 65.5% agreed that difficulty in understanding statistical terms hinders EBM application, with a net agreement of 60.2%.

## Attitude toward EBM

Most pharmacists (83.3%) agreed that practicing EBM improves patient outcomes, with a high net agreement of 81.4%. Additionally, 80.0% of pharmacists expressed willingness to learn or practice EBM (net agreement of 79.6%). A majority (78.8%) of the pharmacists believed that EBM enhances work effectiveness, and 78.6% felt that it is essential for pharmacists to update their EBM knowledge continually. However, 51.2% of the pharmacists perceived EBM as a potential threat to good clinical practice (net agreement 24.3%). There was mixed sentiment on experience versus EBM, with 40.3% favoring experience (net agreement 18%). Moreover, 57.7% believed understanding basic disease mechanisms suffices for good practice (net agreement of 38.3%), and 50.0% felt reading systematic review conclusions is adequate (net agreement of 28.7%). Further details on pharmacists' attitudes toward EBM are in Table 3.

## Practice of EBM

The data show that 51.4% of pharmacists frequently or very frequently apply EBM, with a net frequency of 37%. A high proportion (85.5%) of pharmacists reported using multiple search engines for systematic reviews, with a net frequency of 72.4%. However, time constraints hinder EBM practice, with 34.0% of pharmacists frequently lacking time to study or apply it (net frequency 15.6%). Continuous medical education on EBM was frequently engaged by 40.7% of pharmacists (net frequency 17.9%), and 43.6% frequently shared EBM knowledge with colleagues (net frequency 26.1%), indicating a positive trend in evidence-based culture. Additionally, 41.7% of the pharmacists frequently used the PICO format to translate clinical questions, with a net frequency of 20.9%, suggesting moderate adoption of this EBM skill. Further statistics on pharmacists' EBM practices are in Table 4.

**Table 3. Pharmacists' attitude toward EBM.**

| Statement | SA/A | N | SD/D | Positive responses | Negative responses | Net agreement |
|---|---|---|---|---|---|---|
| I believe that EBM is a threat to good clinical practice | 107 | 44 | 55 | 51.2% | 26.9% | 24.3% |
| Practicing EBM can improve patient health outcomes | 171 | 31 | 4 | 83.3% | 1.9% | 81.4% |
| I am willing to learn/practice EBM if given the opportunity. | 166 | 39 | 1 | 80.0% | 0.4% | 79.6% |
| I feel that research findings are very important in my day-to-day management of patients | 161 | 40 | 5 | 78.1% | 2.4% | 75.7% |
| I believe years of experience are more valuable than EBM | 83 | 77 | 46 | 40.3% | 22.3% | 18% |
| I am convinced that applying EBM in practice increases the effectiveness of my work | 162 | 38 | 6 | 78.8% | 2.9% | 75.9% |
| I am confident that understanding the basic mechanisms of disease is sufficient for good clinical practice | 119 | 47 | 40 | 57.7% | 19.4% | 38.3% |
| I feel that access to databases is vital in obtaining journals on EBM | 138 | 62 | 7 | 66.9% | 3.4& | 63.5% |
| I feel that reading the conclusions of a systematic review is adequate for clinical practice | 103 | 59 | 44 | 50.0% | 21.3% | 28.7% |
| I think it is mandatory to continuously update pharmacists' knowledge/education in EBM to deliver efficient patient care | 162 | 39 | 5 | 78.6% | 2.4% | 76.2% |
| Total Score | Means (%) | | | | General Impression | |
| 7827/10300 | 38 (76) | | | | Neutral Attitude | |

SD = Strongly Disagree, D = Disagree, N = Neutral, A = Agree, SA = Strongly Agree;

Net Agreement = (% Agree + % Strongly Agree) - (% Strongly Disagree + % Disagree).

**Table 4. Pharmacists' practice of EBM.**

| Statement | F/VF | O | R/N | High Frequency | Low Frequency | Net Frequency |
|---|---|---|---|---|---|---|
| I apply EBM in practice | 106 | 71 | 29 | 51.4% | 14.0% | 37% |
| I use multiple search engines for systematic reviews | 95 | 81 | 27 | 85.5% | 13.1% | 72.4% |
| I do not have enough time to study/practice EBM | 70 | 98 | 38 | 34.0% | 18.4% | 15.6% |
| I join Continuous Medical Education for an update regarding EBM | 84 | 75 | 47 | 40.7% | 22.8% | 17.9% |
| I promote and share knowledge about EBM with my colleagues at the workplace | 90 | 80 | 36 | 43.6% | 17.5% | 26.1% |
| I usually translate a clinical question into a form that can be answered from the literature (PICO)[a] | 86 | 77 | 43 | 41.7% | 20.8% | 20.9% |
| Total Score | Means (%) | | | General Impression | | |
| 4211/6180 | 21.2 (68) | | | Fair Practice | | |

[a]PICO; P = Patient or Population; I = Intervention or Indicator; C = Comparison or Control; O = Outcome; T = Time or Type. N = Never, R = Rarely, O = Occasionally, F = Frequently, VF = Very Frequently. High. Net Frequency = (% Frequently + % Very Frequently) - (% Never + % Rarely).

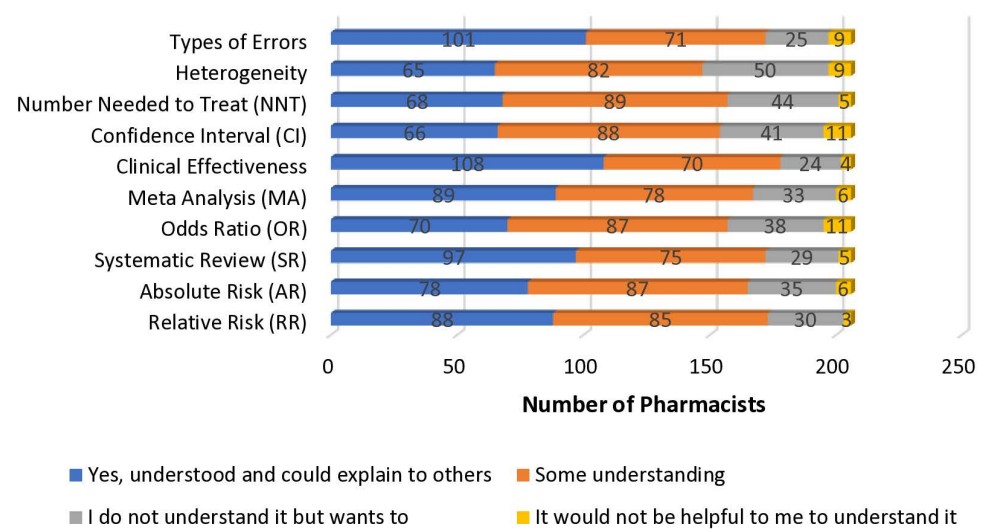

**Fig 1. Pharmacists' awareness of technical terms used in EMB.**

## Statistical terms in EMB

The survey analysis revealed that pharmacists had varied levels of understanding of statistical terms. Accordingly, 52.4% found clinical effectiveness to be the most understandable, whereas heterogeneity was found to be challenging, with 24% lacking any understanding. Additionally, 5% of pharmacists felt that understanding confidence intervals (CI) would not benefit them, while 18.4% showed interest in learning more about CIs despite limited understanding. 18.4% reported not understanding the odds ratio, 17.0% did not comprehend absolute risk, and 15.0% lacked an understanding of relative risk. Furthermore, only 33% of pharmacists were able to understand and explain "the number needed to treat." Fig 1 summarizes pharmacists' responses, highlighting their comprehension and interest in further learning of EBM-related technical terms.

**Table 5. Association between pharmacists' demographic characteristics and knowledge, attitude, and practice.**

| Demographic Characteristics | Knowledge score | | Attitude Score | | Practice score | |
|---|---|---|---|---|---|---|
| | Mean ± SD | p-value | Mean ± SD | p-value | Mean ± SD | p-value |
| **Age** | | 0.35 | | 0.22 | | 0.72 |
| 20-30 | 3.75 ±0.52 | | 3.83 ±0.49 | | 3.85 ±0.48 | |
| 31-40 | 3.77 ±0.57 | | 3.76 ±0.53 | | 3.79 ±0.52 | |
| 41-50 | 4.01 ±0.62 | | 3.81 ±0.71 | | 4.02 ±0.74 | |
| 51-60 | – | | - | | – | |
| **Gender** | | 0.82 | | 0.34 | | 0.98 |
| Male | 3.77 ±0.56 | | 3.83 ±0.52 | | 3.83 ±0.53 | |
| Female | 3.78 ±0.52 | | 3.74 ±0.49 | | 3.83 ±0.54 | |
| **Practice Settings** | | 0.03* | | 0.60 | | 0.02* |
| Hospital Pharmacists | 3.84 ±0.52 | | 3.83 ±0.50 | | 3.89 ±0.51 | |
| Community Pharmacists | 3.70 ±0.57 | | 3.76 ±0.53 | | 3.76 ±0.58 | |
| **Level of Education** | | 0.35 | | 0.84 | | 0.87 |
| Bachelor's Degree in Pharmacy (B-Pharm) | 3.81 ±0.58 | | 3.83 ±0.545 | | 3.81 ±0.54 | |
| Bachelor's Degree in Doctor of Pharmacy (PharmD) | 3.80 ±0.59 | | 3.79 ±0.54 | | 3.86 ±0.53 | |
| Postgraduate (MSc/PhD) in Pharmacy | 3.90 ±0.46 | | 3.72 ±0.57 | | 3.84 ±0.55 | |
| **Source of Education** | | 0.54 | | 0.63 | | 0.09 |
| Saudi University | 3.78 ±0.54 | | 3.81 ±0.53 | | 3.86 ±0.53 | |
| Foreign University | 3.75 ±0.57 | | 3.76 ±0.50 | | 3.76 ±0.54 | |
| **Years of Experience** | | 0.05 | | 0.74 | | 0.01* |
| < 1 | 3.88 ± 0.53 | | 3.97 ±0.51 | | 4.12 ±0.45 | |
| 1–5 | 3.69 ±0.50 | | 3.71 ±0.47 | | 3.78 ±0.57 | |
| 6–10 | 3.65 ±0.47 | | 3.78 ±0.44 | | 3.68±0.42 | |
| >10 | 3.78 ±0.63 | | 3.86 ±0.60 | | 3.90 ±0.58 | |
| **Attended an Education/Training Program on EBM** | | 0.04* | | 0.03* | | 0.75 |
| Yes | 3.85 ±0.50 | | 3.89 ±0.47 | | 3.83 ±0.48 | |
| No | 3.68 ±0.55 | | 3.69 ±0.54 | | 3.83 ±0.58 | |

*Significant p-value

## Association between pharmacists' demographics and KAP scores

Table 5 explores the association between pharmacists' demographic characteristics and their KAP scores related to EBM. The analysis of KAP scores between hospital and community pharmacists revealed significant differences in knowledge and practice domains. Hospital pharmacists exhibited significantly higher knowledge scores (M = 3.84, SD = 0.52) than community pharmacists (M = 3.70, SD = 0.57), with a p-value of 0.03 and an effect size of 0.20. No significant difference was found between the two groups in attitude scores. In the practice domain, hospital pharmacists again outperformed community pharmacists (M = 3.89, SD = 0.51 vs. M = 3.76, SD = 0.58), with a p-value of 0.03 and an effect size of 0.22. Both effect sizes indicate a small effect according to Cohen's classification.

The study also found that years of experience were significantly associated with practice scores (p = 0.01), with pharmacists having less than one year of experience reporting higher practice scores than those with 1–5 or 6–10 years of experience. Additionally, attending EBM training programs was associated with slightly higher knowledge (p = 0.04, effect size = 0.19) and practice scores (p = 0.03, effect size = 0.20). Both effect sizes indicate a small effect according to Cohen's classification.

## Discussion

This study provides valuable insights into the KAP of EBM among hospital and community pharmacists in the Madinah region of Saudi Arabia. Our research extends previous work by providing the first direct comparison of EBM implementation between practice settings, revealing significant differences in adoption patterns. Building upon earlier hospital-focused studies [13], the study findings demonstrate the impact of practice environment on EBM implementation, with hospital pharmacists showing higher knowledge scores compared to community pharmacists. The findings reveal a generally positive attitude toward EBM but highlight areas for improvement and potential barriers to its implementation. The results indicate a moderate to high level of knowledge about EBM principles among pharmacists. The majority of pharmacists (84.4%) correctly identified EBM as involving the critical appraisal of research for clinical decision-making, aligning with Tebala's seminal definition of EBM [33]. However, there was some confusion regarding the role of clinical experience in EBM, with 41.7% of the participants believing that EBM focuses solely on research evidence. This misconception suggests a need for clarification on the integration of research evidence with clinical expertise and patient values, as Wieten emphasized in their discussion of EBM principles [34]. The high agreement (81.1%) that EBM improves the understanding of research questions and methodology is encouraging, as it suggests that pharmacists recognize the value of EBM when enhancing their professional skills. This finding is consistent with those of Landey and Sibbld, who reported similar perceptions among healthcare professionals [35].

The study revealed generally positive attitudes toward EBM, with 83.3% agreeing that EBM can improve patient health outcomes. This is consistent with the findings of Abu Farha et al., who reported positive attitudes toward EBM among Jordanian pharmacists [9]. The high willingness to learn and practice EBM (80.0%) indicates a receptive environment for further EBM education and implementation initiatives. However, the perception of EBM as a potential threat to good clinical practice by 51.2% of respondents is concerning. This could stem from misunderstandings about EBM's role in clinical decision-making or concerns about the devaluation of clinical experience. Similar concerns have been noted in other studies, such as those by Haynes et al., highlighting the need for education that emphasizes EBM as a complement to, rather than a replacement for, clinical expertise [36].

The EBM practice among pharmacists showed room for improvement. Although 51.4% frequently reported applying EBM in practice, only 41.7% regularly used the PICO format to translate clinical questions. This gap between knowledge and practice is common and has been observed in other healthcare settings [37].

The high frequency of systematic reviews on multiple search engines (85.5%) is encouraging, suggesting that pharmacists actively seek evidence. However, the reported time constraints (34.0% frequently lacking time for EBM) repeat findings from other studies and highlight a significant barrier to EBM implementation [13].

The study identified significant differences between hospital and community pharmacists regarding their EBM-related knowledge and practice. Hospital pharmacists demonstrated marginally higher scores, which may be attributed to the structured hospital environment that fosters continuous professional development and collaboration with other healthcare professionals [38]. However, the lower EBM scores among community pharmacists cannot be explained solely by differences in practice settings.

Several additional factors likely contribute to this disparity. Community pharmacists often face limited access to evidence-based resources, lack formal clinical support systems, and experience greater time constraints due to higher workload demands [39,40]. Unlike hospital pharmacists, who typically work within multidisciplinary teams and benefit from access to institutional guidelines and electronic databases, community pharmacists may lack the infrastructure necessary to support regular engagement with EBM practices [41].

To bridge this gap, targeted interventions are needed. Improving access to clinical resources, reducing workload burdens, and providing structured EBM training tailored to community pharmacy settings could significantly enhance the uptake of EBM [42,43]. Strengthening these areas may help support the broader and more consistent application of EBM practices across both hospital and community pharmacy sectors.

 

The influence of experience on EBM adoption reveals an interesting paradox: pharmacists with less than one year of experience scored higher in knowledge, attitudes, and practice (KAP) compared to their more experienced counterparts. One possible explanation is that recent graduates may have received more formal and updated training in evidence-based medicine (EBM) due to changes in pharmacy curricula. In contrast, pharmacists with more years of experience may not have had the same level of EBM emphasis during their initial training, which may explain the lower practice scores among this group [44–46]. Another contributing factor could be that younger pharmacists generally show greater interest in applying EBM compared to older pharmacists, as previous research suggests that both age and years of experience can influence EBM implementation in practice [47]. This trend is consistent with findings in other health professions, where integrated EBM curricula have been shown to significantly improve information literacy and evidence application skills among students and recent graduates [48].

The positive impact of EBM training on knowledge and attitude scores, but not on practice scores, suggests a gap between theoretical understanding and practical application. Studies have indicated that although healthcare professionals often demonstrate improved knowledge and positive attitudes following EBM training, the translation of this knowledge into practice remains insufficient due to barriers such as time constraints, lack of access to resources, and insufficient integration of EBM into daily routines [19,49].

The study also identified a significant barrier: 41.7% of pharmacists mistakenly believed that EBM disregards clinical experience. This misconception reflects a fundamental misunderstanding of EBM's core principle, which emphasizes the integration of clinical expertise, the best available research evidence, and patient values. Therefore, addressing this misunderstanding is essential to promote a more accurate and balanced perception of EBM among pharmacists and to support its broader adoption in practice.

## Recommendations for enhancing EBM adoption among pharmacists

Overcoming the barriers to EBM adoption identified in this study requires a comprehensive, multi-faceted approach that addresses both individual competencies and systemic support structures. One recommended approach is the implementation of blended learning models that combine face-to-face workshops with online modules, focusing on key competencies such as evidence appraisal, database searching, and the application of research findings to clinical scenarios [50]. Furthermore, short, targeted workshops and flexible, self-paced online courses could better accommodate the busy schedules of practicing pharmacists. Mentorship programs, in particular, offer an additional layer of continuous, real-world support, helping pharmacists to translate EBM principles into their daily routines [51,52]. Beyond individual training, institutional support plays a vital role in facilitating EBM adoption. Organizations can embed clinical decision-support tools into pharmacists' workflows, provide protected time specifically for EBM-related activities, and encourage interdisciplinary collaboration. By implementing these multi-faceted strategies, institutions can help build both the confidence and competency required for pharmacists to consistently apply EBM in everyday practice [52–55].

## Limitations of the study

Several limitations should be considered when interpreting the results of this study. First, the study design, which utilized Google Forms' branching logic to exclude diploma-holding pharmacy technicians, non-participants, and incomplete responses, prevented the collection of comparative data, thereby limiting the ability to assess selection bias. Second, the achieved response rate of 38.15% and the relatively young average age of participants may impact the representativeness of the findings. Third, the focus on pharmacists within the Madinah region, coupled with the use of a convenience sampling method, restricts the generalizability of the results to other regions of Saudi Arabia. Additionally, the self-administered nature of the questionnaire may have introduced social desirability bias, potentially leading to an overestimation of EBM knowledge and practices. Finally, the cross-sectional study design precludes the assessment of temporal changes in EBM adoption patterns over time.

## Conclusion

This study reveals significant variations in Evidence-Based Medicine (EBM) implementation between hospital and community pharmacists in the Madinah region, with hospital pharmacists demonstrating slightly higher knowledge and practice scores. Despite generally positive attitudes toward EBM, with 83.3% believing it improves patient outcomes, significant barriers persist, including time constraints and insufficient training. The findings suggest that EBM education positively influences knowledge and practice, though its impact on attitudes remains limited. These results highlight the need for targeted interventions, particularly for community pharmacists, focusing on practical EBM application and addressing workplace-specific barriers. Future initiatives should emphasize continuous professional development, workplace support systems, and integrated EBM training programs that bridge the gap between theoretical knowledge and practical application.

## Acknowledgments

The authors of this study acknowledge the contribution of the pharmacists who participated in this study.

## Author contributions

**Conceptualization:** Fahad Alzahrani, Nawaf Almutairi, Abdullah Aloufi, Naif Alarawi, Abdulmalik Kattan, Abdulaziz Hakeem, Mohammed Alharbi.

**Data curation:** Fahad Alzahrani, Nawaf Almutairi, Abdullah Aloufi, Naif Alarawi, Abdulmalik Kattan, Abdulaziz Hakeem, Mohammed Alharbi.

**Formal analysis:** Fahad Alzahrani, Ehsan Habeeb.

**Investigation:** Fahad Alzahrani, Nawaf Almutairi, Abdullah Aloufi, Naif Alarawi, Abdulmalik Kattan, Abdulaziz Hakeem, Mohammed Alharbi.

**Methodology:** Fahad Alzahrani, Nawaf Almutairi, Abdullah Aloufi, Naif Alarawi, Abdulmalik Kattan, Abdulaziz Hakeem, Mohammed Alharbi.

**Project administration:** Fahad Alzahrani, Nawaf Almutairi, Abdullah Aloufi, Naif Alarawi, Abdulmalik Kattan.

**Resources:** Fahad Alzahrani.

**Software:** Fahad Alzahrani, Ehsan Habeeb.

**Supervision:** Fahad Alzahrani, Nawaf Almutairi.

**Validation:** Fahad Alzahrani, Nawaf Almutairi.

**Visualization:** Fahad Alzahrani.

**Writing – original draft:** Fahad Alzahrani, Nawaf Almutairi, Abdullah Aloufi, Naif Alarawi, Abdulmalik Kattan, Abdulaziz Hakeem, Mohammed Alharbi, Ehsan Habeeb, Haifa A. Fadil.

**Writing – review & editing:** Fahad Alzahrani, Nawaf Almutairi, Abdullah Aloufi, Naif Alarawi, Abdulmalik Kattan, Abdulaziz Hakeem, Mohammed Alharbi, Ehsan Habeeb, Haifa A. Fadil.

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
