## [Decision Letter · Decision Letter 0]

16 Jan 2025

PONE-D-24-39283Adoption of Evidence-Based Medicine: A Comparative Study of Hospital and Community Pharmacists in Saudi ArabiaPLOS ONE

Dear Dr. Alzahrani,

Thank you for submitting your manuscript to PLOS ONE. After careful consideration, we feel that it has merit but does not fully meet PLOS ONE’s publication criteria as it currently stands. Therefore, we invite you to submit a revised version of the manuscript that addresses the points raised during the review process.

As the editor, I would like to offer the following general comments:

Introduction: Please ensure that your introduction is scientifically sound. Begin with a general overview and then narrow down to specific details. Be sure to incorporate existing solutions to the problem, highlight the research gap, and reference previous studies.

Language Editing: I recommend enhancing the overall English language quality of your manuscript.

Abstract: Please revise your abstract to ensure it is more compelling and meets the journal's standards.

Thank you for your attention to these matters.

We look forward to receiving your revised manuscript.

Kind regards,

Habtamu Setegn Ngusie

Academic Editor

PLOS ONE

Please confirm at this time whether or not your submission contains all raw data required to replicate the results of your study. Authors must share the “minimal data set” for their submission. PLOS defines the minimal data set to consist of the data required to replicate all study findings reported in the article, as well as related metadata and methods (https://journals.plos.org/plosone/s/data-availability#loc-minimal-data-set-definition ).

If your submission does not contain these data, please either upload them as Supporting Information files or deposit them to a stable, public repository and provide us with the relevant URLs, DOIs, or accession numbers. For a list of recommended repositories, please see https://journals.plos.org/plosone/s/recommended-repositories .

Reviewers' comments:

Reviewer's Responses to Questions

**Comments to the Author**

1. Is the manuscript technically sound, and do the data support the conclusions?

Reviewer #1: Yes

Reviewer #2: Partly

2. Has the statistical analysis been performed appropriately and rigorously? 

Reviewer #1: Yes

Reviewer #2: Yes

3. Have the authors made all data underlying the findings in their manuscript fully available?

Reviewer #1: Yes

Reviewer #2: Yes

4. Is the manuscript presented in an intelligible fashion and written in standard English?

Reviewer #1: Yes

Reviewer #2: Yes

5. Review Comments to the Author

Reviewer #1: Actuality, Evidence Based Medicine has become an essential discipline for the decision-making process in the individual care of patients, with a judicious, and conscious use of the best scientific available evidence.

The relevance of this study takes root in the need to incorporate in an accurate way the basic knowledge of EBM into daily pharmacist’s practice. There are still few studies that evaluate the knowledge, attitudes and practices of pharmacists regarding the practice of EBM.

I consider that the document is written appropriately, presents a clear methodology, with adequate data analysis and presentation of results.

I don’t have any observation or consideration regarding the document.

Reviewer #2: The study employed convenience sampling and a combination of purposive and snowball sampling methods to select research subjects, which might lead to selection bias among respondents. Additionally, only 38.1% of the recruited samples were ultimately included in the analysis, thereby limiting the validity and representativeness of the results. Therefore, it is recommended to supplement the analysis with a comparative study of some of the unincluded samples to further clarify the authenticity and representativeness of the results.

6. PLOS authors have the option to publish the peer review history of their article (what does this mean? ). If published, this will include your full peer review and any attached files.

**Do you want your identity to be public for this peer review?** For information about this choice, including consent withdrawal, please see our Privacy Policy .

Reviewer #1: No

Reviewer #2: No

---

## [Author Response · Author response to Decision Letter 1]

20 Jan 2025

Editor general comments:

i. Comment: Introduction: Please ensure that your introduction is scientifically sound. Begin with a general overview and then narrow down to specific details. Be sure to incorporate existing solutions to the problem, highlight the research gap, and reference previous studies.

Response: Thank you for your valuable feedback. We have thoroughly revised the introduction to make sure of scientific soundness and proper scholarly progression. Comment: Language Editing: I recommend enhancing the overall English language quality of your manuscript.

Response: Thank you for your valuable comment. We have carefully revised the manuscript to enhance its clarity, precision, and academic tone. Here are the specific improvements made.

ii. Comment: Abstract: Please revise your abstract to ensure it is more compelling and meets the journal’s standards.

Response: Thank you for your feedback. We have thoroughly revised the abstract to make it more compelling and aligned with journal standards.

Reviewer's Responses

Reviewer #1 Comments: Actuality, evidence-based medicine has become an essential discipline for the decision-making process in the individual care of patients, with judicious and conscious use of the best scientific evidence available.

The relevance of this study takes root in the need to incorporate in an accurate way the basic knowledge of EBM into daily pharmacist’s practice. There are still few studies that evaluate the knowledge, attitudes, and practices of pharmacists regarding the practice of EBM.

Responses: We appreciate your insights and have strengthened our manuscript to emphasize these important points better. First, we enhanced the introduction to establish the study's significance better. Second, we have strengthened the justification for the research gap.

Reviewer #2 comment: The study employed convenience sampling and a combination of purposive and snowball sampling methods to select research subjects, which might lead to selection bias among respondents. Additionally, only 38.1% of the recruited samples were ultimately included in the analysis, thereby limiting the validity and representativeness of the results. Therefore, it is recommended to supplement the analysis with a comparative study of some of the unincluded samples further to clarify the authenticity and representativeness of the results.

Response: We appreciate your concerns about the sampling methodology. This study used two sampling techniques: convenience and snowball sampling. We acknowledge that our initial manuscript incorrectly included purposive samplingand have removed this reference. The text in the manuscript has been updated.

Regarding selection bias although convenience sampling can introduce selection bias, our choice was informed by both practical considerations and methodological precedent in clinical and social settings.[1-3]. We have addressed this limitation by explicitly acknowledging it in our limitations section.

Concerning the suggestion to conduct a comparative analysis with unincluded samples, we must note that the majority of excluded responses were due to incomplete data, specifically where participants discontinued upon reaching questions about their educational qualifications.  

References

1. Stratton SJ. Population research: convenience sampling strategies. Prehospital and disaster Medicine. 2021;36(4):373-4.

2. Simkus J. Convenience sampling: Definition, method and examples. Retrieved Oktober. 2022;6:2022.

3. Etikan I, Musa SA, Alkassim RS. Comparison of convenience sampling and purposive sampling. American journal of theoretical and applied statistics. 2016;5(1):1-4.

---

## [Decision Letter · Decision Letter 1]

26 Mar 2025

PONE-D-24-39283R1Adoption of Evidence-Based Medicine: A Comparative Study of Hospital and Community Pharmacists in Saudi ArabiaPLOS ONE

Dear Dr. Alzahrani,

Thank you for submitting your manuscript to PLOS ONE. After careful consideration, we feel that it has merit but does not fully meet PLOS ONE’s publication criteria as it currently stands. Therefore, we invite you to submit a revised version of the manuscript that addresses the points raised during the review process.

**Authors are required to reply all the queries, raised by the reviewers.** Please submit your revised manuscript by May 10 2025 11:59PM. If you will need more time than this to complete your revisions, please reply to this message or contact the journal office at plosone@plos.org . Please include the following items when submitting your revised manuscript:

We look forward to receiving your revised manuscript.

Kind regards,

Priti Chaudhary, M.S.

Academic Editor

PLOS ONE

Reviewers' comments:

Reviewer's Responses to Questions

**Comments to the Author**

1. If the authors have adequately addressed your comments raised in a previous round of review and you feel that this manuscript is now acceptable for publication, you may indicate that here to bypass the “Comments to the Author” section, enter your conflict of interest statement in the “Confidential to Editor” section, and submit your "Accept" recommendation.

Reviewer #2: (No Response)

Reviewer #3: (No Response)

2. Is the manuscript technically sound, and do the data support the conclusions?

Reviewer #2: Partly

Reviewer #3: Partly

3. Has the statistical analysis been performed appropriately and rigorously? 

Reviewer #2: Yes

Reviewer #3: Yes

4. Have the authors made all data underlying the findings in their manuscript fully available?

Reviewer #2: No

Reviewer #3: Yes

5. Is the manuscript presented in an intelligible fashion and written in standard English?

Reviewer #2: Yes

Reviewer #3: Yes

6. Review Comments to the Author

**Reviewer #2:**  1) The response rate after sampling in this study was only 38.15%, and the average age of the participants was relatively young (31 years old), so the representativeness of the research results is limited and there may be selective bias. It is suggested to supplement the overall number of research subjects that meet the inclusion and exclusion criteria, as well as compare the basic information of the sample obtained from this sampling with the overall population.

2) In addition, the research topic is a comparison between hospital and community pharmacists, and there is a lack of corresponding KAP comparison data in the results. Suggest adding baseline data comparison between hospitals and community pharmacists; Comparison of KAP survey results between hospital and community pharmacists.

**Reviewer #3:**  The manuscript titled "Adoption of Evidence-Based Medicine: A Comparative Study of Hospital and Community Pharmacists in Saudi Arabia" is well-structured and presents valuable insights into pharmacists' knowledge, attitudes, and practices regarding EBM. The study is methodologically sound, with clear objectives, a well-defined methodology, and a thorough statistical analysis. The discussion effectively interprets the results, comparing them with existing literature. However, some areas require improvement before recommending the manuscript for publication.

Major recommendations

Clarity and Consistency in Writing

• The manuscript is generally well-written, but some sections contain redundancy. For instance, in the introduction, the definition of EBM is repeated in multiple ways. Consider merging these statements for conciseness.

• Some sentences are overly long and complex, making them harder to follow. For example, in the introduction: "It combines the best available scientific evidence with the clinician's expertise and the patient's values to guide medical decisions effectively." This could be revised for better flow: "EBM integrates scientific evidence, clinical expertise, and patient values to improve medical decision-making."

Research Justification and Novelty

• The study’s importance is well-stated, but there is limited emphasis on the novelty of the research. Consider explicitly stating how this study fills a gap in previous research. For example:

o Does this study provide the first comparative analysis of hospital vs. community pharmacists in Saudi Arabia?

o How does it add to previous research findings on EBM adoption in pharmacy practice?

Strengthening this section will help establish the study's significance.

Methodology

• The response rate (38.15%) is relatively low. Was there any effort to mitigate potential non-response bias?

• The use of convenience and snowball sampling raises concerns about selection bias. While this is acknowledged, a stronger justification is needed to explain why this method was chosen over random sampling.

• The exclusion criteria are mentioned briefly. Were there any specific reasons for excluding pharmacists from the pharmaceutical industry or academia? Clarifying this will strengthen the study's rigor.

Statistical Analysis - Additional Clarifications

• The manuscript states that Bloom's threshold was used to categorize knowledge, attitude, and practice scores. While this is a reasonable approach, it would be beneficial to justify why this method was chosen over other classification methods.

• The use of net agreement and net frequency scores is well-documented, but some readers may not be familiar with these terms. Consider adding a brief explanation or reference in the Methods section.

• The effect size (Cohen’s d = 0.15 - 0.16) is interpreted as "small." This should be explicitly stated in the Results section, as some readers may not be familiar with effect size thresholds.

Interpretation of Findings

• The results indicate that pharmacists with less than one year of experience had higher EBM practice scores than more experienced pharmacists. This is an interesting finding but needs further exploration. Could it be that recent graduates receive better EBM training? If so, how can this be addressed in continuing education programs?

• Community pharmacists had lower EBM scores than hospital pharmacists. The manuscript attributes this to differences in professional settings, but other factors (e.g., lack of access to resources, workload differences) should be discussed in more depth.

• The misconception that EBM ignores clinical experience was reported by 41.7% of participants. This is a crucial barrier and should be highlighted in the discussion as a key area for intervention.

Practical Implications and Recommendations

• The manuscript does a good job of discussing barriers to EBM adoption, but the recommendations for overcoming these barriers are somewhat vague.

o What specific training strategies should be implemented?

o Would workshops, online courses, or mentorship programs be more effective?

o How can institutions support pharmacists in overcoming time constraints?

Adding specific recommendations will strengthen the manuscript's practical impact.

Minor Changes

Abstract:

o “EBM in Saudi Arabia’s pharmacies is an under-researched area despite its importance in pharmacy practice.”

o Consider rewording to: "Despite its importance in pharmacy practice, EBM adoption in Saudi Arabian pharmacies remains under-researched."

7. PLOS authors have the option to publish the peer review history of their article (what does this mean? ). If published, this will include your full peer review and any attached files.

**Do you want your identity to be public for this peer review?** For information about this choice, including consent withdrawal, please see our Privacy Policy .

Reviewer #2: No

Reviewer #3: No

---

## [Author Response · Author response to Decision Letter 2]

26 Apr 2025

Reviewer's Responses

Reviewer #2

1. Comments: The response rate after sampling in this study was only 38.15%, and the average age of the participants was relatively young (31 years old), so the representativeness of the research results is limited and there may be selective bias. It is suggested to supplement the overall number of research subjects that meet the inclusion and exclusion criteria, as well as compare the basic information of the sample obtained from this sampling with the overall population.

Responses: Thank you for this valuable comment. We acknowledge that the response rate of 38.15% and the relatively young average age of participants may limit the representativeness of our findings and introduce potential selection bias. We have addressed this limitation in the revised manuscript under the "Study Limitations" section.

Regarding the suggestion to supplement the number of research subjects and compare the basic characteristics of the sample with the overall population, we appreciate the recommendation. However, due to the design of our online survey using Google Forms' branching logic, we automatically excluded pharmacy technicians holding diploma degrees, a vital segment of the healthcare workforce in Saudi Arabia, along with individuals who declined participation or submitted incomplete surveys. The survey system immediately terminated responses for diploma holders after the education level question, and incomplete surveys were not stored in our database. Similarly, we have no demographic or comparative data for non-respondents. Therefore, while we recognize this as a significant limitation, we are unable to perform the suggested comparison between respondents and non-respondents or analyze potential differences between complete and incomplete survey responses. We have highlighted this limitation and recommended that future research use stratified sampling and larger, more inclusive datasets to enhance representativeness and allow for more detailed comparisons.

2. Comment: In addition, the research topic is a comparison between hospital and community pharmacists, and there is a lack of corresponding KAP comparison data in the results. Suggest adding baseline data comparison between hospitals and community pharmacists; Comparison of KAP survey results between hospital and community pharmacists.

Response: Thank you for your constructive feedback. We have addressed your concerns as follows:

1. Baseline Data Comparison:

We have expanded Table 1 to include detailed baseline demographic characteristics comparing hospital and community pharmacists.

2. KAP Survey Results Comparison:

We have enhanced our results section, which provides a detailed comparison of KAP scores between hospital and community pharmacists as follows:

a. Knowledge Domain:

Hospital pharmacists demonstrated significantly higher knowledge scores (M = 3.84, SD = 0.52) compared to community pharmacists (M = 3.70, SD = 0.57), with statistical significance (p = 0.03).

b. Attitude Domain:

No significant difference was observed in attitude scores between hospital pharmacists (M = 3.83, SD = 0.50) and community pharmacists (M = 3.76, SD = 0.53; p = 0.60).

c. Practice Domain:

Hospital pharmacists showed significantly higher practice scores (M = 3.89, SD = 0.51) compared to community pharmacists (M = 3.76, SD = 0.58; p = 0.02).

Reviewer #3

Clarity and Consistency in Writing

1. Comment: The manuscript is generally well-written, but some sections contain redundancy. For instance, in the introduction, the definition of EBM is repeated in multiple ways. Consider merging these statements for conciseness.

Response: Thank you for your comment. We have revised the introduction to be more concise by removing redundant definitions of EBM and streamlining the content.

2. Comment: Some sentences are overly long and complex, making them harder to follow. For example, in the introduction, "It combines the best available scientific evidence with the clinician's expertise and the patient's values to guide medical decisions effectively." This could be revised for better flow: "EBM integrates scientific evidence, clinical expertise, and patient values to improve medical decision-making."

Response: Thank you for your feedback. We have revised the sentence as suggested.

Research Justification and Novelty

3. Comment: The study’s importance is well-stated, but there is limited emphasis on the novelty of the research. Consider explicitly stating how this study fills a gap in previous research. For example:

a. Does this study provide the first comparative analysis of hospital vs. community pharmacists in Saudi Arabia?

b. How does it add to previous research findings on EBM adoption in pharmacy practice?

Strengthening this section will help establish the study's significance.

Response: We appreciate your feedback. Thank you for your input. We have revised our manuscript to emphasize the significance of our study, particularly highlighting its status as the first comparative analysis of EBM adoption between hospitals and community pharmacists in Saudi Arabia, in both the concluding paragraph of the introduction and the opening section of the discussion. In addition, we have revised the discussion section to illustrate what our study adds to previous research.

Methodology

4. Comment: The response rate (38.15%) is relatively low. Was there any effort to mitigate potential non-response bias?

Response: Thank you for your valuable comment. While our response rate of 38.15% is lower than ideal, it aligns with similar pharmacy practice surveys conducted in Saudi Arabia. For instance, Al-Jazairi and Alharbi reported a 20% response rate among hospital pharmacists when assessing evidence-based practice [1]. Another cross-sectional study targeting hospital pharmacists yielded a response rate of 14.8% [2]. Additionally, the “National Survey of Drug Information Centers Practice in Saudi Arabia” reported a 40% response rate, involving pharmacists in drug information centers, which may include community pharmacists [3]. This limitation has been explicitly addressed in the revised manuscript, highlighting its potential implications for the generalizability and interpretation of the study findings.

5. Comment: The use of convenience and snowball sampling raises concerns about selection bias. While this is acknowledged, a stronger justification is needed to explain why this method was chosen over random sampling.

Response: While random sampling would be ideal, convenience and snowball sampling were employed due to several context-specific challenges. First, there was no comprehensive, accessible sampling frame of pharmacists in Saudi Arabia. For instance, Fathelrahman conducted a nationwide cross-sectional study assessing medical device-related counseling practices among community pharmacists in Saudi Arabia [4]. They employed convenience sampling due to the lack of a national registry or contact database for pharmacists, which prevented the use of probability sampling methods and stratification by geographical region. This limitation underscores the challenges in obtaining a representative sample of pharmacists in specific regions like the Madinah region of Saudi Arabia. Consequently, our study utilized convenience and snowball sampling techniques to reach pharmacists across diverse practice settings, particularly community pharmacists who are typically harder to access through institutional channels. While this approach may introduce selection bias, it aligns with established practices in pharmacy research where complete listings of the target population are unavailable. The Study Sampling and Response Rate section has been updated to reflect this rationale.

6. Comment: The exclusion criteria are mentioned briefly. Were there any specific reasons for excluding pharmacists from the pharmaceutical industry or academia? Clarifying this will strengthen the study's rigor.

Response: Thank you for your observation. Our study specifically focused on hospital and community pharmacists, as they are the primary healthcare professionals actively involved in patient care and clinical decision-making informed by evidence-based medicine (EBM). This choice is supported by prior research indicating that these pharmacists are key users of EBM in daily practice, particularly in medication therapy management, clinical interventions, and patient-centered care [5, 6]. Pharmacists working in industry and academia were excluded, as they typically do not hold active patient care licenses and are not routinely engaged in clinical decision-making that necessitates EBM application. This exclusion aligns with the methodology of similar studies investigating EBM use among frontline healthcare providers, which have similarly omitted participants from non-clinical sectors [7-9]. By targeting licensed pharmacists involved in direct patient care, our study offers a more accurate reflection of EBM implementation in routine pharmacy practice. We have clarified this rationale in the revised Study Population subsection of the Methods section to enhance transparency and methodological rigor.

Statistical Analysis - Additional Clarifications

7. Comment: The manuscript states that Bloom's threshold was used to categorize knowledge, attitude, and practice scores. While this is a reasonable approach, it would be beneficial to justify why this method was chosen over other classification methods.

Response: Thank you for this thoughtful comment. The use of Bloom's threshold in categorizing knowledge, attitude, and practice (KAP) scores is a common methodological approach in cross-sectional studies, as demonstrated by multiple sources, particularly in the absence of standardized benchmarks [10-13]. While none of the provided studies explicitly justify why Bloom’s cut-off was selected over other classification methods, their consistent application of this method supports its validity as a standardized tool in KAP research. We have added a brief justification in the Methods section to clarify the rationale for this choice.

Comment: The use of net agreement and net frequency scores is well-documented, but some readers may not be familiar with these terms. Consider adding a brief explanation or reference in the Methods section.

Response: Thank you for your helpful comment. We have added a short explanation and supporting reference in the Methods section to define these measures and their relevance in KAP studies.

8. Comment: The effect size (Cohen’s d = 0.15 - 0.16) is interpreted as "small." This should be explicitly stated in the Results section, as some readers may not be familiar with effect size thresholds.

Response: Thank you for the comment. Cohen’s d is a commonly used measure of effect size that indicates the standardized difference between two means. According to Cohen’s (1988) guidelines, an effect size of 0.2 is considered small, 0.5 is medium, and 0.8 or above is large. A small effect (d ≈ 0.2) suggests a modest difference between groups that, while statistically significant, may have limited practical significance [14]. We have revised the Methods and Results sections to clearly state that the observed effect sizes fall within the small range.

Interpretation of Findings

9. Comment: The results indicate that pharmacists with less than one year of experience had higher EBM practice scores than more experienced pharmacists. This is an interesting finding but needs further exploration. Could it be that recent graduates receive better EBM training? If so, how can this be addressed in continuing education programs?

Response: Thank you for this thoughtful and insightful comment. We agree that the finding warrants further exploration. One possible explanation is that recent graduates may have received more formal and updated training in evidence-based medicine (EBM) as part of evolving pharmacy curricula [14-16]. In contrast, pharmacists with more years of experience may not have had the same level of EBM emphasis during their initial training, which may explain the lower practice scores among this group. This trend is consistent with findings in other health professions, where integrated EBM curricula have been shown to significantly improve information literacy and evidence application skills among students and recent graduates [17]. This highlights a potential gap in ongoing professional development and underscores the importance of integrating targeted EBM content into continuing education and professional development programs. We have expanded the Discussion section to include this interpretation and its implications.

10. Comment: Community pharmacists had lower EBM scores than hospital pharmacists. The manuscript attributes this to differences in professional settings, but other factors (e.g., lack of access to resources, workload differences) should be discussed in more depth.

Response: Thank you for your comment. We do agree that the observed differences in EBM scores between community and hospital pharmacists may be influenced by a broader range of factors beyond the practice setting alone. Community pharmacists often face barriers such as limited access to clinical resources, lack of institutional support, and higher workload and time constraints, which may hinder their ability to engage with EBM regularly. In contrast, hospital pharmacists may have better access to multidisciplinary teams, structured protocols, and clinical decision-support tools. We have expanded the Discussion section to reflect these considerations and provide a more comprehensive interpretation of the findings.

11. Comment: The misconception that EBM ignores clinical experience was reported by 41.7% of participants. This is a crucial barrier and should be highlighted in the discussion as a key area for intervention.

Response: Thank you for your observation. We have emphasized this point in the revised Discussion section and identified it as a key area for educational and institutional intervention.

Practical Implications and Recommendations

12. Comment: The manuscript does a good job of discussing barriers to EBM adoption, but the recommendations for overcoming these barriers are somewhat vague.

a. What specific training strategies should be implemented?

b. Would workshops, online courses, or mentorship programs be more effective?

c. How can institutions support pharmacists in overcoming time constraints?

Response: We appreciate your comment. We have expanded the Discussion section to include evidence-based strategies for overcoming key barriers to EBM adoption. For training, structured continuing education programs that combine interactive workshops, case-based learning, and evidence retrieval exercises are recommended. Online courses and webinars offer flexibility and broad accessibility, especially for community pharmacists. Mentorship programs led by experienced clinical pharmacists can also provide personalized guidance and reinforce practical application. To address time constraints, institutions can support EBM practice by integrating clinical decision support tools into pharmacy workflows, allocating protected time for professional development, and fostering a culture that values evidence-based care. These enhancements have been added to the revised manuscript to provide a more comprehensive roadmap for improving EBM uptake.

Minor Changes

Abstract:

13. Comment: EBM in Saudi Arabia’s pharmacies is an under-researched area despite its importance in pharmacy practice.” Consider rewording to: "Despite its importance in pharmacy practice, EBM adoption in Saudi Arabian pharmacies remains under-researched."

Response: Thank you for the suggestion. The sentence has been updated to: “Despite its importance in pharmacy practice, EBM adoption in Saudi Arabian pharmacies remains under-researched.”

References

1. Al-Jazairi AS, Alharbi R. Assessment of evidence-based practice among hospital pharmacists in Saudi Arabia: attitude, awareness, and practice. International Journal of Clinical Pharmacy. 2017;39:712-21.

2. Ajabnoor AM, Cooper RJ. Pharmacists’ prescribing in Saudi Arabia: cross-sectional study describing current practices and future perspectives. Pharmacy. 2020;8(3):160.

3. Alomi Y, Alghamdi S, Alattyh R. National Survey of Drug

---

## [Editor Report · Decision Letter 2]

29 Apr 2025

Adoption of Evidence-Based Medicine: A Comparative Study of Hospital and Community Pharmacists in Saudi Arabia

PONE-D-24-39283R2

Dear Dr. Fahad Alzahrani,

We’re pleased to inform you that your manuscript has been judged scientifically suitable for publication and will be formally accepted for publication once it meets all outstanding technical requirements.

Kind regards,

Priti Chaudhary, M.S.

Academic Editor

PLOS ONE
---

## [Editor Report · Acceptance letter]

PONE-D-24-39283R2

PLOS ONE

Dear Dr. Alzahrani,

I'm pleased to inform you that your manuscript has been deemed suitable for publication in PLOS ONE. Congratulations! Your manuscript is now being handed over to our production team.

Kind regards,

on behalf of

Dr. Priti Chaudhary

Academic Editor

PLOS ONE